# Optimal Design of THEDES Based on Perillyl Alcohol and Ibuprofen

**DOI:** 10.3390/pharmaceutics12111121

**Published:** 2020-11-20

**Authors:** Eduardo Silva, Filipe Oliveira, Joana M. Silva, Ana Matias, Rui L. Reis, Ana Rita C. Duarte

**Affiliations:** 13B’s Research Group, I3Bs-Research Institute on Biomaterials, Biodegradables and Biomimetics, University of Minho, Headquarters of the European Institute of Excellence on Tissue Engineering and Regenerative Medicine, Avepark, Zona Industrial da Gandra, 4805-017 Barco GMR, Portugal; eduardo.silva@i3bs.uminho.pt (E.S.); rgreis@i3bs.uminho.pt (R.L.R.); 2ICVS/3B’s PT Government Associated Laboratory, University of Minho, 4805-017 Guimarães, Portugal; 3LAQV-REQUIMTE, Chemistry Department, Faculty of Science and Technology, Nova University of Lisbon, 2829-516 Caparica, Portugal; fsn.oliveira@campus.fct.unl.pt; 4Nutraceuticals and Bioactives Process Technology Laboratory, Instituto de Biologia Experimental e Tecnológica, 2780-157 Oeiras, Portugal; amatias@ibet.pt

**Keywords:** therapeutic deep eutectic systems, green chemistry, antimicrobial properties, natural molecules, active pharmaceutical ingredients, anticancer properties

## Abstract

Therapeutic deep eutectic systems (THEDES) have dramatically expanded their popularity in the pharmaceutical field due to their ability to increase active pharmaceutical ingredients (APIs) bioavailability. However, their biological performance has not yet been carefully scrutinized. Herein, THEDES based on the binary mixture of perillyl alcohol (POH) and ibuprofen (IBU) were prepared using different molar ratios. Our comprehensive strategy includes the characterization of their thermal and structural behavior to identify the molar ratios that successfully form deep eutectic systems. The in vitro solubility of the different systems prepared has demonstrated that, unlike other reported examples, the presence of the terpene did not affect the solubility of the anti-inflammatory agent in a physiological simulated media. The biological performance of the systems was studied in terms of their antimicrobial activity against a wide panel of microorganisms. The examined THEDES showed relevant antimicrobial activity against all tested microbial strains, with the exception of *P. aeruginosa*. A synergistic effect from the combination of POH and IBU as a eutectic system was verified. Furthermore, the cytotoxic profile of these eutectic systems towards colorectal cancer (CRC) in vitro cell models was also evaluated. The results provide the indication that the cell viability varies in a dose-dependent manner, with a selective THEDES action towards CRC cells. With tunable bioactivities in a ratio-dependent manner, THEDES enhanced the antimicrobial and anticancer properties, representing a possible alternative to conventional therapies. Therefore, this study provides foreseeable indications about the utility of THEDES based on POH and IBU as strong candidates for novel active pharmaceutical systems.

## 1. Introduction

Since ancient times, compounds of natural origin have played an essential role in society, either as inspiration for various novel active pharmaceutical ingredients (APIs) or as therapeutic agents themselves [1,2]. Among their different sources, medicinal herbs and plants have proven to be essential in the development of modern medicine. Currently, as a result of centuries of compiling information about these molecules, we now have a vast library of promising raw material, with a broad spectrum of chemically diverse structures and bioactivities [1,3,4,5]. Considering the ongoing pursue for sustainability and compliance with the green chemistry metrics, natural compounds have experienced a revised interest not only in academia, but also in industry, revealing their essential contribution to solving the challenges faced by modern medicine [3,6,7,8].

Over the years, with pharmacology refinement and the emergence of several efficient fractionation and separation techniques, several plant-derived classes of compounds started earning attention from the scientific community due to their remarkable biological properties [1,3,5,6,8,9]. Terpenes represent a prime example of these compounds. They are a diverse class of organic compounds produced by several plants and some particular insects, with interesting bioactivities [10,11,12,13]. Terpenes and terpenoids (i.e., terpenes with some extent of chemical modifications) are constituents of the majority of essential oils produced from plants and flowers. These compounds possess several uses in various fields, from cosmetics and food to pharmaceutical and biotechnology industries. As pharmaceutical ingredients, terpenes present several beneficial effects, namely, as antioxidant, anti-inflammatory, anti-carcinogenic and antibacterial agents [13,14,15,16]. Limonene, a cyclic monoterpene, and the major component of the oil produced from citrus fruits peels, is a good example of the versatility displayed by these compounds, since it finds application in all the above-mentioned fields [17,18]. Limonene has been used as a common dietary supplement, as a botanical insecticide, in tissue preparation for histology and as a solvent for cleaning, among other applications [17,19,20,21]. Its bioactivity is also quite remarkable, possessing both significant anti-carcinogenic and antimicrobial activity, which are mainly attributed to its metabolites, such as perillic acid (PA) [22,23,24,25,26]. In this sense, it has been reported that limonene works as a sort of pro-drug since the therapeutic effect of its metabolites has been shown to be far greater [18,25,27]. Aside from PA, perillyl alcohol (POH) is another interesting limonene metabolite that has shown tremendous therapeutic potential [28]. This naturally occurring terpene is produced as a secondary metabolite in plants, via the mevalonate pathway, by hydroxylation of its precursor limonene. POH possesses several enticing biological properties, such as significant antimicrobial potential against a number of clinically relevant pathogenic organisms (e.g., *Candida albicans*, *Fusobacterium nucleatum* and *Porphyromonas gingivalis*) [29,30,31]. However, while POH has shown antitumoral activity both in vitro and in vivo, it has failed to produce satisfactory results in a human clinical trial setting [28,32,33]. In fact, one of the main problems of POH application has been the administration form, which is in part a consequence of its poor water solubility and resulting poor bioavailability. Considering that carcinogenesis is a phenomenon not only restricted to abnormal cell proliferation, and angiogenesis and inflammation play an important role in tumor progression, it is expected that the combination of an anticancer agent, such as POH, with a well-described, non-steroidal anti-inflammatory drug (NSAID) might result in a eutectic entity with enhanced anticancer properties. In this context, ibuprofen is an ideal candidate being one of the most widely applied NSAIDs in clinical practice with a known association with tumor reduction. Furthermore, recent studies have highlighted the potential of deep eutectic systems containing ibuprofen as an API for cancer treatment [34,35].

Firstly, reported by Abbot and coworkers in 2004, deep eutectic systems (DES) have emerged as a competitive “green” alternative to organic solvents and ionic liquids (ILs) [36,37,38]. DES can be defined as a mixture of two or more compounds that in a specific molar ratio present a significant reduction in the system melting point in comparison with its individual compounds. This depression is majorly attributed to the establishment of hydrogen-bond interactions between the compounds, although electrostatic interactions and Van der Waals forces can also play an important role [38,39,40,41]. When one of the system components is an API, it is hence called a therapeutic deep eutectic system (THEDES) [39,42,43]. Their potential goes way beyond being a mere solvent for chemical engineering processes. DES already have extensive applications in the pharmaceutical and biomedical fields [36,38,41,44], being extremely interesting as eutecticity has shown the ability to tune the solubility and permeation of several compounds, including various APIs [35,39].

Hence, the main goal of this study was the development of a THEDES combining POH and ibuprofen (IBU), to improve their solubility and bioavailability, creating a new potential therapeutic formulation displaying two main biological effects, namely, being anti-bacterial and anti-cancer (Figure 1).

## 2. Materials and Methods

THEDES preparation: THEDES were prepared by mixing (S)-(−)-POH (W266418, Sigma Aldrich, St. Louis, MO, USA) and IBU (I4883, Sigma Aldrich, St. Louis, MO, USA) at different molar ratios. The systems were mixed under constant stirring at 60 °C. After 30 min, a clear solution was obtained, and the THEDES were left to cool down at room temperature (RT).

Differential scanning calorimetry (DSC): Experiments were performed, for each formulation, using a TA instrument, the DSC Q100 model (TA instruments, New Castle, DE, USA), in an aluminum pan. The program used consisted of an initial heating step from −20 °C to 100 °C (heating rate 5 °C min^−1^), followed by an isothermal step of 2 min prior to a final cooling step at 20 °C. Experiments were performed under a nitrogen atmosphere (purge gas flux of ca. 50 mL min^−1^).

Nuclear magnetic resonance (NMR): NMR experiments were carried out with a 400 MHz Bruker Advance II. Mestrenova 12.0 software (Mestrelab Research, Santiago, Spain) was used for spectral processing. The THEDES and raw materials were dissolved (30 mg/mL) in dimethyl sulfoxide-d6 (DMSO-d6, 99.9 atom % D, LOT. STBH4385, Sigma Aldrich). All the experiments were performed when the systems were in equilibrium and no further change in their properties was observed.

Solubility evaluation: The solubility assessment was performed using the pure API and the previously prepared THEDES. Briefly, an excess of IBU and THEDES were added to a phosphate-buffered saline solution (PBS, Sigma Aldrich, St. Louis, MO, USA), in separate vials, at 37°, and stirred for 72 h. Thereafter, the samples were filtered using a hydrophilic PTFE syringe filter with a 0.22 µm pore size (Filter Lab, Barcelona, Spain). The determination of IBU solubility was performed by HPLC, using a Thermo Scientific Finnigan Surveyor (Thermo Scientific, Waltham, MA, USA), equipped with a quaternary pump, solvent degasser, auto sampler and column oven, coupled to a UV-VIS detector (Accela, Thermos Scientific, USA). The column used was a keystone kromasil 5 μm particle size, pore size 100 Å, L × I.D. 250 mm × 4.6 mm (Thermo Scientific, Waltham, MA, USA) and the column temperature was 30 °C. The chromatographic separation was performed using a mobile phase consisting of PBS (pH 6.8): acetonitrile = 65:35, *v/v*. The injection volume was 10 µL with a flow rate of 0.7 mL/min, following the procedure described by Jahan et al. [45] with the absorbance of the solutions measured at 222 nm. A calibration curve using the respective THEDES and the API as standards was prepared for quantification.

Assessment of antimicrobial activity: The antimicrobial potential of THEDES was evaluated against microorganisms of clinical relevance, namely, *Staphylococcus aureus* (ATCC 25923), an *S. aureus* methicillin-resistant strain (MRSA) (ATCC 700698), an *S. epidermis* methicillin-resistant strain (MRSE) (ATCC 35984), *Pseudomonas aeruginosa* (ATCC 27853), *Escherichia coli* (ATCC 25922) and *Candida albicans* (ATCC 90029). The assessment was carried out as described in a previous work via a two-step methodology [43,46]. Briefly, the formulations are first subjected to a disk diffusion assay (DDA), using loaded blank antibiotic discs (CT0998B, Oxford), followed by the Minimum Inhibitory Concentration (MIC), Minimum Bactericidal Concentration (MBC) and Minimum Fungicidal Concentration (MFC) determination for the formulations that show relevant activity in DDA. For comparison purposes, the isolated compounds were included in both steps, as internal controls, and suitable antibiotics were used as the positive controls for antibacterial activity. The effective concentration range used for MIC/MBC/MFC determination was between 2500 and 156.25 μg/mL. Experiments were always carried out in triplicate, using independent microbial cultures to account for biological variance. It should be noted that only formulations that retained a liquid state at room temperature were used to carry out these assays.

Assessment of cytotoxicity, antiproliferative effect and selectivity index determination: THEDES were evaluated in terms of their cytotoxicity and antiproliferative effects. The cytotoxic effect, here working as a preliminary safety indicator, was assessed using a continuous cell line culture of heterogeneous human epithelial colorectal adenocarcinoma cells (Caco-2) (ACC 169, DSMZ, Braunschweig, Germany). [47]. Briefly, the cells were subcultured in RPMI medium (Corning, Corning, NY, USA), supplemented with 10% heat-inactivated fetal bovine serum (FBS, Corning, USA) and a 1% penicillin–streptomycin solution (PS, Corning, NY, USA). The cell culture was maintained in a humidified atmosphere at 37 °C with 5% CO_2_. The cytotoxicity assay was performed in accordance with ISO/EN 10993 guidelines. Caco-2 cells were seeded into 96-well plates at a density of 2 × 10^4^ cells/well and allowed to grow for 7 days, with medium renewal every 48 h. At Day 7, cells were incubated with culture media (control) and different THEDES concentrations diluted in culture medium. After 24 h, cells were washed twice with PBS and the cell viability was assessed using a CellTiter 96^®^ AQueous One Solution Cell Proliferation Assay (Promega, Madison, WI, USA), containing an MTS (3-(4,5-dimethylthiazol-2-yl)-5-(3-carboxymethoxyphenyl)-2-(4-sulfophenyl)-2H-tetrazolium) viability reagent. Briefly, 100 μL of the viability reagent was added at each well in a 1:10 dilution and incubated for 3 h. The absorbance was measured at 490 nm using a microplate reader (VICTOR NivoTM, PerkinElmer, Waltham, MA, USA) and cell viability was expressed in terms of percentage of living cells relative to the control. Three independent experiments were performed in triplicate.

The antiproliferative effect towards cancer cells was assessed using a continuous cell culture of human Caucasian colon adenocarcinoma (HT29) (ACC 299, DSMZ, Germany); these cells form a well-differentiated colorectal adenocarcinoma (CRC), and thus have been accepted as a CRC cell model in 2D and 3D in vitro cultures. The cells were subcultured as described above. Briefly, HT29 cells were seeded at a density of 1 × 10^5^ cells/well in 96-well culture plates. After 24 h, cells were incubated with culture media (control) and with different THEDES concentrations diluted in culture medium. Cell proliferation was measured after 24 h using the MTS viability reagent, as previously described. Furthermore, THEDES selectivity indexes were calculated as a ratio between the half maximal effective concentrations (EC_50_) from the cytotoxic and antiproliferative profiles previously obtained.

## 3. Results

### 3.1. Design and Characterization of THEDES

The design of THEDES is still a trial-and-error process due to the lack of knowledge on the interactions established between the counterparts and, thereby, computational simulation programs. Table 1 summarizes the visual aspects of the different formulations at RT, indicating that up to a molar ratio of 2:1 pasty-like solids were obtained, whereas using a higher amount of POH (i.e., molar ratio 3:1) led to a transparent liquid with a few crystal particles on the bottom of the vials. Using molar ratios of 4:1; 6:1 and 8:1, no crystals were detected, and a completely transparent liquid was successfully obtained. The liquid phase is a strong indication of lack of lattice arrangement and intermolecular interactions between the counterparts, as elsewhere reported [39,42,48].

The THEDES and IBU powder were analyzed by DSC up to 100 °C (Figure 2) in order to evaluate the thermal events and confirm the results obtained by simply naked-eye observation. Compared with the IBU (≈77.7 °C) thermogram, a shift of the melting point in THEDES was observed, which indicates a gradual depression on the melting point of IBU as the mole fraction of POH increases. The thermograms of THEDES up to a molar ratio of 3:1 present an endothermic peak, ranging from ≈62.2 °C to ≈38.3 °C, which is far lower than the melting point of IBU.

Owing to the gap in understanding the interactions between the compounds, the supramolecular arrangement of THEDES were further evaluated using HNMR, allowing to study the intermolecular interactions of the atoms involved and even to confirm the ratios of the counterparts. The HNMR spectra for the isolated components and the molar ratios that remained in a liquid state at RT are presented in Figure 3 and Figure 4, respectively. The remaining spectra are included in the Appendix A.

There is a clear change in the peak correspondent to the POH hydroxyl group (δ = 4.59) (Figure 3A) between the isolated compound and the compound while in THEDES form. Namely, the well-defined triplet verified for the isolated compounds is presented, generally, when in THEDES form as a larger singlet (Figure 4), which is a hallmark of hydrogen-bond interactions. Likewise, it is possible to observe a suppression of the peak correspondent to IBU’s OH (δ = 12.21) (Figure 3B) when in THEDES form (Figure 4). These two facts heavily suggest the establishment of interactions between the two compounds, via hydrogen bonds, corroborating the results obtained by DSC analysis. 1H–1H-nuclear Overhauser enhancement (NOESY) spectroscopy was also performed, with the results included as Appendix A.

### 3.2. Solubility of Ibuprofen in Physiologically Simulated Media

In this work, we have evaluated the solubility of ibuprofen in the liquid samples prepared, to understand how the effect of the increase in terpene ratio could be related to the solubility enhancement of ibuprofen. The solubility evaluation was performed in POH:IBU molar ratios of 3:1, 4:1, 6:1 and 8:1, in comparison with IBU alone, but also POH and IBU in a physical mixture (POH + IBU), where the two individual components are dissolved in PBS using the same concentration as in the THEDES. In opposition to the previously reported results, an enhancement of IBU solubility could not be identified when using this eutectic formulation (Figure 5).

### 3.3. Antimicrobial Potential of the Formulated THEDES

To measure the antimicrobial effect of the developed THEDES, a two-step methodology was applied. Firstly, a DDA was performed against a panel of microbial organisms relevant at the clinical level. It should be noted that antimicrobial activity was only measured for the systems that were fully liquid at RT, namely, a POH:IBU of 4:1, 6:1 and 8:1. The obtained results are presented in Table 2.

As can be easily seen by analyzing the obtained inhibition halo measurements, both the isolated POH and all the tested THEDES display relevant antimicrobial activity against the selected panel of microorganisms, with the exception of *P. aeruginosa*, showcasing the efficacy of these formulations against not only Gram-positive bacteria and *C. albicans*, a fungus with a “Gram-positive-like” membrane structure, but also against Gram-negative bacteria, such as *E. coli*. Considering the results obtained in the DDA, all the THEDES formulations were selected for posterior studies to determine the MIC/MBC/MFC concentrations for the microorganisms that displayed susceptibility to the compounds. The results obtained are presented on Table 3 and Table 4.

The obtained concentration values confirm the previously obtained results, which further highlights the systems’ efficacy against both bacteria types as well as fungi, with the obtained MIC/MBC/MFC values being concurrent with the inhibition halo measurements. In order to understand these observations, the absolute mass composition of each formulation was determined and is presented in Figure 6.

### 3.4. THEDES Anticancer Activity

POH chemoprotective and anticancer properties, namely against colorectal cancer, have been widely described [49]. Anti-inflammatory drugs can play an important role in angiogenesis and inflammation processes and, therefore, have an important role in tumor progression [50]. Herein, the anticancer activity of the most promising THEDES was explored combining both the individual compounds with interesting bioactivities. For this purpose, the eutectic system that was fully liquid at RT and had the most promising antimicrobial activity, POH:IBU 8:1 and POH:IBU 3:1, the system with the lowest molar ratio liquid at 37 °C was considered. Furthermore, the effect of POH and IBU as a physical mixture (POH + IBU) was also investigated, aiming to explore the contrast between both components as a eutectic entity and as an aqueous solution. The obtained results are presented in Table 5 and highlighted in Figure 7.

## 4. Discussion

In this work, new THEDES formulations were produced by combining a non-steroidal anti-inflammatory drug (i.e., IBU) with POH, which is a penetration enhancer that also has anticancer properties [52,53,54,55]. POH was mixed with IBU at equimolar and imbalanced molar ratios to determine the formulations that would be subjected to further analysis (Table 1). The simplicity of production is of paramount importance for the pharmaceutical field as THEDES yields are 100% with no need for further purification steps [56,57,58]. The DSC analysis of the systems prepared has shown that by further increasing the mole fraction of POH, a complete suppression on the melting point ascribed to IBU was achieved, suggesting the loss of lattice arrangement. The depression of the melting point is highly dependent on the starting components, molar ratio, lattice energy, intermolecular interactions and entropy [48,59,60]. Similar data was obtained for the other THEDES systems, being a plausible reason for the establishment of intermolecular interactions between the counterparts and the supramolecular arrangement of the counterparts while in THEDES form [39,42,43]. Overall, the DSC data are in good agreement with the naked-eye observations, being also aligned with previous data reported in the literature for the THEDES based on limonene:IBU [35].

Furthermore, NMR studies confirmed the successful production of the THEDES. Analyzing the obtained spectra, it is possible to verify the purity of the isolated compounds, since the spectra presented in Figure 3 are in accordance with previous reports in the literature [61,62]. Additionally, it is possible to confirm the accuracy of the molar ratios for the various formulations, as the peak integral values of the THEDES spectra (Figure 4) vary in accordance to the proportion of the formulation’s components. In general, it is possible to verify the existence of intermolecular interactions between the THEDES components.

The solubility of an API is often a limiting factor for its bioavailability, with consequences in its therapeutic effectiveness [63]. In previous studies, it has been reported that eutectic mixtures are able to increase an API’s solubility [42,64]. NSAIDs such as ibuprofen present very low solubility in water (≈21 mg/L) [65], but in combination with menthol as an eutectic mixture experiences an solubility enhancement of 12.76-fold [57]. In another example, the solubility of ibuprofen combined with limonene in a molar ratio of 1:4 was enhanced 4.3-fold. When the ratio of limonene increases to 8, the solubility enhancement is of the order of 5.63, comparing to the pure form of ibuprofen [35]. The results obtained in this work do not follow this enhancement trend since the combination of POH and IBU as a eutectic mixture does not result in an increase in the API’s solubility when compared with IBU in the powder form (Figure 5).

The antimicrobial potential of the formulated THEDES was evaluated according to two methodologies. In the first approach, DDA demonstrates that the obtained results are in accordance with previous accounts in the literature, as terpenoid-derived compounds have well-documented antimicrobial activity against Gram-positive bacteria and fungi, with some activity against the Gram-negative ones although at quite a lower extent [33,66,67,68]. This phenomenon is usually attributed to the difference in membrane composition/complexity, as Gram-negative bacteria have more complex membrane structures, with multiple accounts in the literature reporting the presence of lipopolysaccharides on the cell wall, which hamper the hydrophobic compounds from reaching the cell and exerting their effects [69,70,71,72]. Regarding the complete lack of inhibition for *P. aeruginosa*, this is most likely due to the well-reported terpene bioconversion capabilities of the *Pseudomonas* genus, which have been in several accounts regarded as one of the most promising biocatalysts for the obtention of terpene-derived products [73,74,75]. As for the complete lack of inhibition observed for IBU, this is most likely a consequence of the compound’s insolubility in aqueous media and consequent incapability to properly diffuse from the disk into the agar media. After this assessment, the MIC/MBC/MFC concentrations for the microorganisms that displayed susceptibility to the compounds were determined. The widespread effect of THEDES on the microorganisms reflects the primary actuation method of terpenes and other essential oil components, namely, cell membrane perturbation, which is non-specific in nature and typically less prone to the development of resistance mechanisms [76,77]. It should be noted that, in the conditions established, it was not possible to dissolve IBU. Furthermore, while this compound is well known for its anti-inflammatory capabilities, some accounts in the literature do report antibacterial activity against certain microorganisms, such as *E. coli* and *Bacillus subtilis* (MIC = 1025–2500 μg/mL) [78,79]. This slight antibacterial activity verified for other organisms may explain why even though MRSE shows greater MIC/MBC values for isolated POH than both *S. aureus* and MRSA, the values obtained for the formulation POH:IBU 8:1 was the same across all the strains. This phenomenon may be the consequence of both the added bioavailability of IBU as well as a possible synergistic effect between the compounds while in THEDES form. This fact has been previously reported in the literature, where it is stated that a THEDES supramolecular arrangement may result in synergist effects, which may result in a more pronounced effect or even reduced toxicity [35,80,81]. Nevertheless, considering the absolute mass composition of these formulations (Figure 6), the major contributor for the verified antimicrobial effect is in all likelihood POH.

Concerning the study on the anticancer activity of the THEDES prepared, the EC_50_ results obtained from the cytotoxicity assay for both systems revealed an increase in the cytotoxic profile towards normal colonic cells (Caco-2 model) as a result of the increase of POH ratio in the eutectic system. These results suggest that the cytotoxic effect of these THEDES is mainly due to the action of the POH itself, as it was expected from what has been reported in the literature [49]. This is also verified when compared to previously reported data on the cytotoxic effect of POH towards this cell line, since the system with a higher molar ratio of POH has a similar cytotoxic profile as POH itself [51]. Although both systems presented similar anticancer effects, with analogous antiproliferative results on colorectal cancer cells (HT29), the POH:IBU (3:1) was the most proficient system to inhibit HT29 proliferation without compromising normal cell viability. This effect may be due to the selective action of the eutectic system on the cancer cell membrane, which induces perturbation and ultimately its disruption, as has been reported for several other DES [82,83]. To better understand the uniqueness of these eutectic entities, the antiproliferative activity of a simple physical mixture of the components was evaluated. Interestingly, the antiproliferative activity revealed to be completely different from both eutectic formulations (Figure 7A,B). These results give the indication that the supramolecular rearrangement promoted by the eutectic formulation allows the preparation of tailor-made systems towards colorectal cancer cells. The selectivity index (SI) of a compound is a parameter used to express a compound’s in vitro efficacy in the inhibition of a particular target. For this matter, it is the result of the ratio between the antiproliferative effect and the cytotoxicity of the eutectic system. Thus, from the SI results we are able to have a preliminary representation of the anticancer potential of the THEDES, alongside with their safety, thereby illustrating a possible therapeutic window for these systems. Therefore, although both systems presented interesting results for the antiproliferative assay, a POH:IBU of 3:1 seems to be the most promising for further establishment of its anticancer activity.

## 5. Conclusions

Herein, POH was for the first time successfully combined with IBU and yielded a THEDES liquid sample when mixed in adequate molar ratios. The obtained data showed that the properties of the parent species do not suffer any negative effects due to their inclusion in a DES supramolecular effect, retaining, generally, the antimicrobial and anticancer proprieties of its major component, POH. In fact, the results obtained in this study point to the existence of synergistic/additive effects resulting from the DES interaction between POH and IBU. The remarkable properties of these THEDES systems mainly lie in the intermolecular interactions between the components, which allow to tailor their properties depending on the molar ratio of the two components present in the system (POH:IBU). The use of THEDES for commercial purposes is still in its infancy; thereby, it is essential to fully characterize them in terms of their physicochemical and biological properties. This is of extreme importance to validate the use of THEDES as therapeutic agents. In this work, the use of THEDES based on POH:IBU has shown to have great potential, both as antimicrobial agent and as anticancer agent. Interestingly, the therapeutic properties of the THEDES designed here are dependent on the molar ratio used, which may lead to selective applications of POH:IBU. The attractive properties displayed by these formulations encourage further studies to understand the detailed mechanisms of action of these systems for the envisioned purposes, namely, infection prevention/treatment and applications in cancer therapy.

## Figures and Tables

**Figure 1 pharmaceutics-12-01121-f001:**
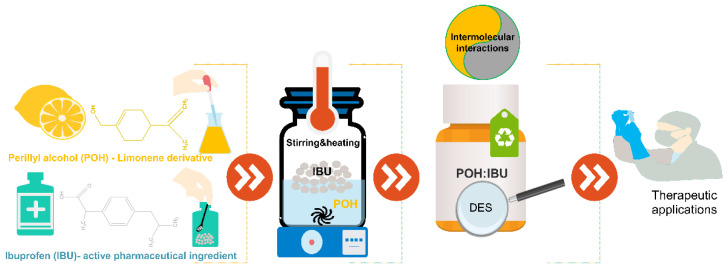
Schematic illustration of the developed perillyl alcohol (POH)-based THEDES.

**Figure 2 pharmaceutics-12-01121-f002:**
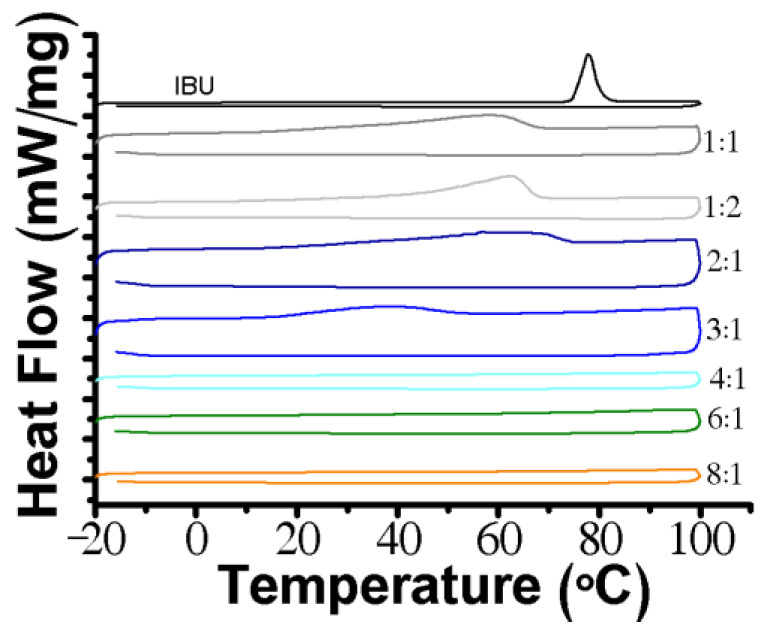
DSC thermogram of IBU and POH:IBU at different molar ratios.

**Figure 3 pharmaceutics-12-01121-f003:**
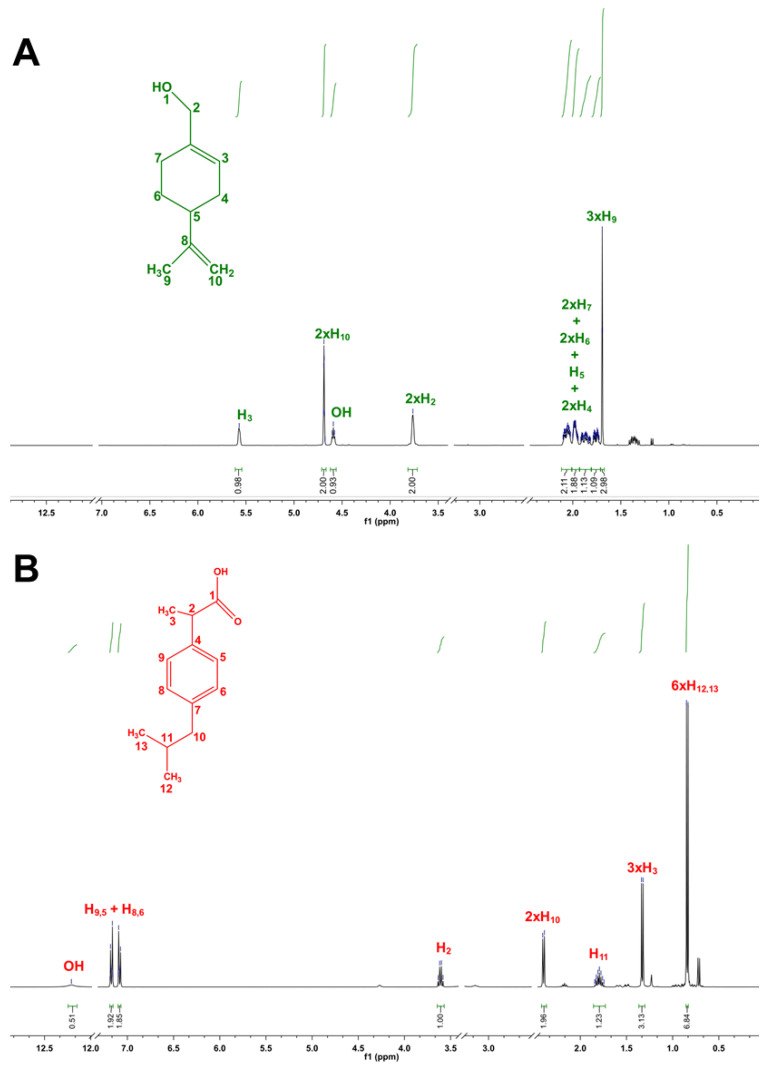
1H NMR spectra of (**A**) POH and (**B**) IBU. Peak assignment and integration were fully performed.

**Figure 4 pharmaceutics-12-01121-f004:**
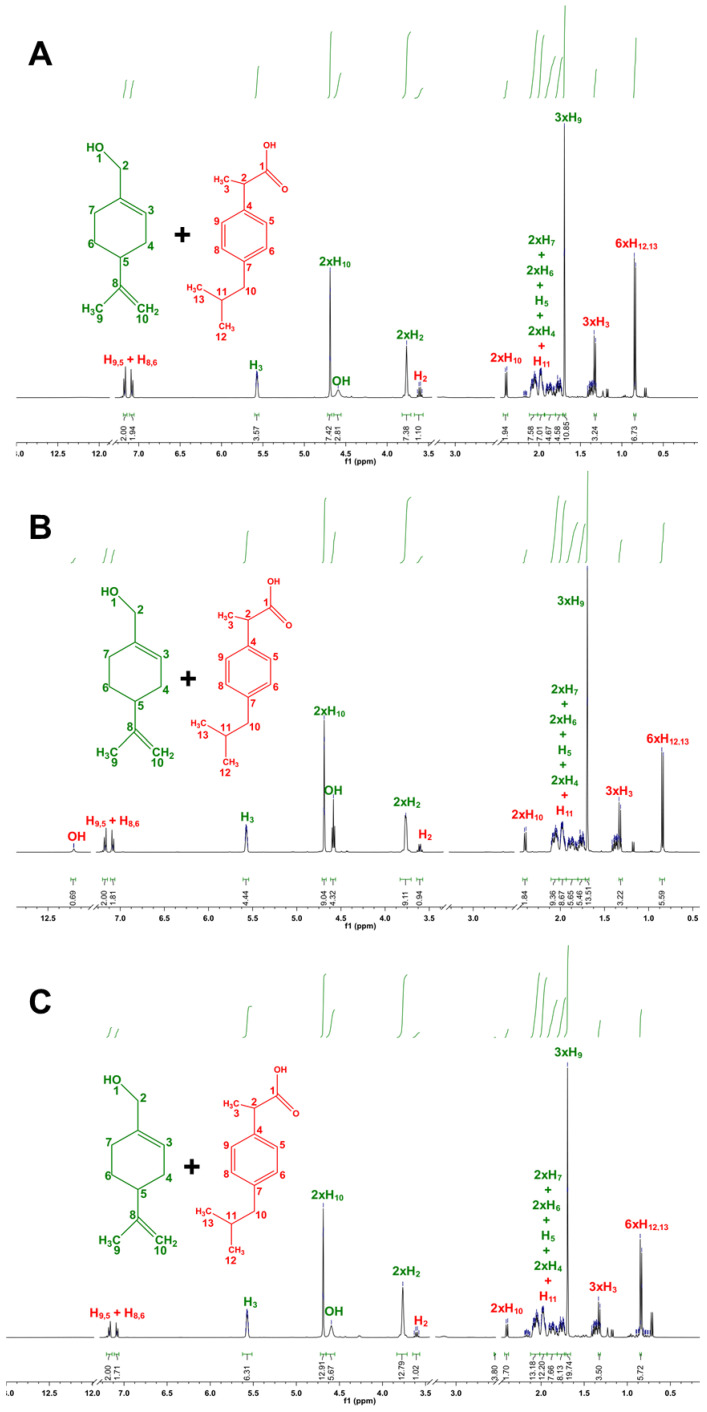
1H NMR spectra of THEDES: (**A**) POH:IBU 4:1, (**B**) POH:IBU 6:1 and (**C**) POH:IBU 8:1. Peak assignment and integration were fully performed.

**Figure 5 pharmaceutics-12-01121-f005:**
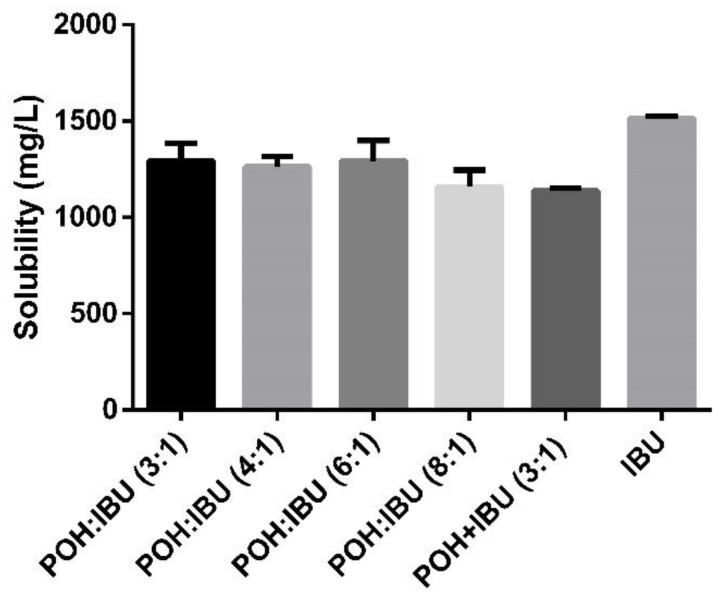
Solubility of IBU in powder form or complexed in THEDES in a PBS solution at physiological-like conditions.

**Figure 6 pharmaceutics-12-01121-f006:**
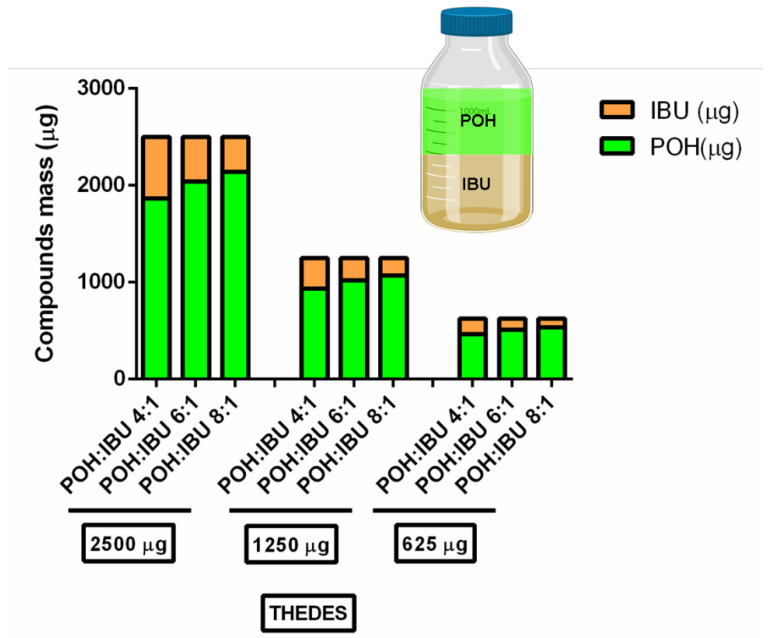
Absolute mass composition of THEDES based on POH and IBU.

**Figure 7 pharmaceutics-12-01121-f007:**
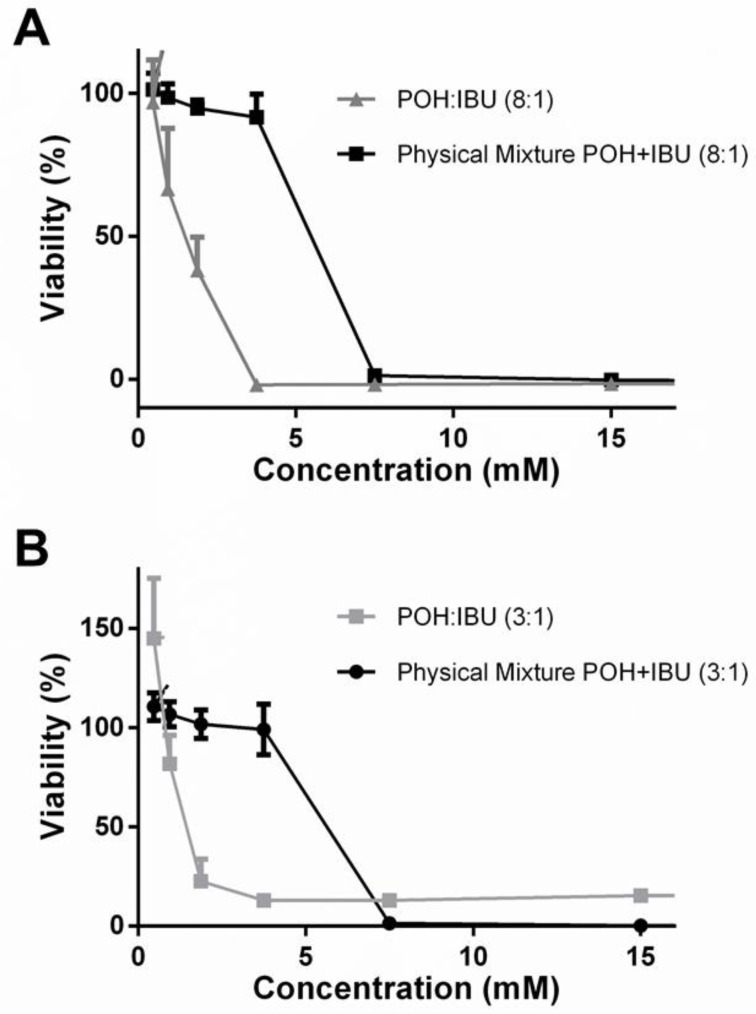
Comparison of the antiproliferative effect of THEDES and a physical mixture of POH and IBU. (**A**) POH:IBU 8:1 and a physical mixture of POH and IBU dissolved in cell culture medium using the same concentration as in the THEDES; (**B**) POH:IBU 3:1 and a physical mixture of POH and IBU dissolved in cell culture medium using the same concentration as in the THEDES. Results were expressed relatively to the control as the mean ± SD of three independent experiments performed in triplicate.

**Table 1 pharmaceutics-12-01121-t001:** Summary of the different POH:IBU therapeutic deep eutectic systems (THEDES) prepared in this study.

Molar Ratio	Visual Aspect at RT	Melting Point (°C)
1:1	Solid	≈57.9
1:2	Solid	≈62.2
2:1	Solid	≈57.9
3:1	Liquid with a few crystals	≈38.3
4:1	Liquid	-
6:1	Liquid	-
8:1	Liquid	-

**Table 2 pharmaceutics-12-01121-t002:** Inhibition halo measurements (diameter (mm) ± SD) for the various THEDES formulations, individual counterparts and controls ^α^. Results are presented by formulation/compound for each microbial strain tested. NI—no inhibition; NT—not tested.

THEDES/Compound	*E. coli*	*P. aeruginosa*	*S. aureus*	MRSA	MRSE	*C. albicans*
POH	17.50 ± 0.41	NI	20.33 ± 0.47	19.17 ± 0.24	21.50 ± 0.70	31.33 ± 1.70
IBU	NI	NI	NI	NI	NI	NI
POH:IBU 4:1	16.67 ± 0.94	NI	21.83 ± 0.24	22.50 ± 0.71 *	24.33 ± 0.85 *	25.60 ± 0.94 *
POH:IBU 6:1	17.33 ± 0.82	NI	19.33 ± 0.41	22.33 ± 2.94 *	24.50 ± 1.97 *	30.67 ± 0.81
POH:IBU 8:1	15.33 ± 0.94	NI	20.33 ± 0.47	21.00 ± 0.41	20.83 ± 0.24	32.00 ± 1.67
Sterile water	NI	NI	NI	NI	NI	NI

^α^ Statistical analysis was performed for all tested THEDES, and the data were considered statistically different for *p* values < 0.05. * Denotes significant differences when compared with the isolated POH.

**Table 3 pharmaceutics-12-01121-t003:** MIC values of the individual counterparts and THEDES. Results are presented by formulation for each microbial strain tested. ND—not dissolved.

	MIC (μg/mL)				
Compound/THEDES	*E. coli*	*S. aureus*	MRSA	MRSE	*C. albicans*
POH	1250	625	625	1250	625
IBU	ND	ND	ND	ND	ND
POH:IBU 4:1	1250	1250	1250	1250	625
POH:IBU 6:1	1250	1250	1250	1250	625
POH:IBU 8:1	1250	625	625	625	625

**Table 4 pharmaceutics-12-01121-t004:** MBC/MFC values of the individual counterparts and THEDES. Results are presented by formulation for each microbial strain tested. ND—not dissolved.

	MBC/MFC (μg/mL)				
Compound/THEDES	*E. coli*	*S. aureus*	MRSA	MRSE	*C. albicans*
POH	2500	1250	1250	2500	1250
IBU	ND	ND	ND	ND	ND
POH:IBU 4:1	2500	2500	2500	2500	1250
POH:IBU 6:1	2500	2500	2500	2500	1250
POH:IBU 8:1	2500	1250	1250	1250	1250

**Table 5 pharmaceutics-12-01121-t005:** EC_50_ values for the cytotoxicity and antiproliferative assays, and obtained selectivity index, for the selected THEDES, individual compounds and physical mixtures. Results were obtained from three independent experiments performed in triplicate.

System/Compound	EC_50_ Values (mM)		Selectivity Index
	Cytotoxicity Assay	Antiproliferative Assay	
POH	4.86 ± 1.58 ^1^	2.37 ± 0.20 ^1^	2.06
IBU	2.89 ± 0.06 ^2^	2.346 ± 0.09 ^2^	1.23
POH:IBU 3:1	8.46 ± 1.13	1.316 ± 0.07	5.89
POH:IBU 8:1	4.35 ± 0.29	1.37 ± 0.09	2.60
POH + IBU 3:1	-	4.51± 0.26	-
POH + IBU 8:1	-	4.43 ± 0.23	-

^1^ Reported by Rodrigues et al. [51]. ^2^ Reported by Pereira et al. [35].

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
