# Peer review of "Optimal Design of THEDES Based on Perillyl Alcohol and Ibuprofen"

_pharmaceutics, 2020, doi:10.3390/pharmaceutics12111121_

Round 1
Reviewer 1 Report
I could not proceed with the reviewing process because there are too many edit histories.
Authors should finalize the manuscript and then, just emphasize the revised points by using color (red letter or somethings).
Also, from the data, I could not understand the reason for adding IBU to POH because from table 2 the antimicrobial potential of developed system is almost same to only POH. What is the main purpose to develop the DES? If the purpose is improvement of solubility and bioavailability of IBU, some additional evaluations focused on IBU should be necessary.
Purpose of the DSC measurement is still unclear.
The interactions between each compound should be evaluated by FT-IR or some other techniques.
In the data of NMR, the range of X-axis should be unified (use same range in each NMR data)
Table 2, what is the difference between NI and 0.00?
Table 5, control sample must be necessary to clarify the benefits of present systems.
Author Response
Reviewer 1
- I could not proceed with the reviewing process because there are too many edit histories. Authors should finalize the manuscript and then, just emphasize the revised points by using colour (red letter or somethings).
The authors would firstly like to apologize for any confusion caused to the reviewer. Due to a reason alien to the authors, the manuscript made available for revision marked the bulk of the modifications made to manuscript as “deleted”. In this letter, the authors will address the reviewer’s concerns point-by-point emphasizing the necessary parts on the revised manuscript. The authors will know address each point in separate stating the Line and nature of the change done in the manuscript (These changes will also be highlighted in the resubmitted manuscript that will accompany this rebuttal letter).
- Also, from the data, I could not understand the reason for adding IBU to POH because from table 2 the antimicrobial potential of developed system is almost same to only POH. What is the main purpose to develop the DES? If the purpose is improvement of solubility and bioavailability of IBU, some additional evaluations focused on IBU should be necessary.
The author would like to, firstly, thank the reviewer for their comment.
In this study the main goal, as stated in the introduction (L96-98), is the design of Perillyl alcohol (POH) and ibuprofen (IBU) based therapeutic deep eutectic solvents (THEDES). Our hypothesis is that the combination of the two molecules in an eutectic system may render a new entity with two main biological activities, namely antibacterial and anticarcinogenic activities. As referred to in the introduction section, POH has documented antibacterial potential and has been extensively studied regarding its anticarcinogenic activities but has failed to reproduce its promising in vitro/ in vivo results in Human clinical trials (L71-75). The choice of IBU has a counterpart to POH stems from its status as one of, if not the most, widely applied non-steroidal anti-inflammatory drug (NSAID). Since inflammation is considered a major player in carcinogenesis and tumour progression, the authors hypothesised that the combination of POH, an anticarcinogenic compound, with a potent anti-inflammatory drug, such as IBU, would results in the enhancement of our formulations capabilities regarding anticarcinogenic potential. In fact, previous reports in the literature already associate IBU with tumour reduction. Finally, there is also some reports, although few in the number, of IBU having some antibacterial potential against bacteria (e.g. Escherichia coli and Bacillus subtilis), fact that is mentioned in the discussion section (L343-345) which also, in part, motivated the choice of IBU for this study.
It is true that the antimicrobial activity is mostly due to the terpene molecule. Nonetheless this results provide us an indication that this system does not present advantages over the individual components. However, in the case of the anticancer activity it is clear the advantage of having a THEDES system (Table 5, Figure 7) Furthermore, the authors would also like to highlight that several accounts in the literature report that compounds in THEDES form display enhanced action. (refer to Results section, point 3.4 – THEDES anticancer activity)
Regarding the purpose of developing a THEDES, the authors agree with the reviewer’s comment as one of the objectives was, in fact, the attempt to improve the bioavailability and solubility of DES. Data regarding this was already included in this resubmission, alas as mentioned above was unfortunately marked as deleted. The added section can be found between L226-237.
- Purpose of the DSC measurement is still unclear.
The author’s present the DSC measurement as it allows a very immediate and visual verification that the formulations developed possess a hallmark trait of DES. Looking at Figure 2, one can easily verify the lowering of melting point of IBU (i.e. gradual displacement of the endothermic peak belonging to this compound and eventual complete suppression). As mentioned by the author in the introduction, DES can be defined as “a mixture of two or more compounds, that in a specific molar ratio, present a significant reduction on the system melting point in comparison with its individual compounds.”. As such the authors believe it to be worthwhile the inclusion of a visual representation of the DSC thermograms in the manuscript.
- The interactions between each compound should be evaluated by FT-IR or some other techniques.
The authors understand and recognized the potential of FT-IR for the inferring of interactions between compounds. While FT-IR could without provide further insights into the intermolecular interactions occurring in the formulation, it is the authors believe that NMR provides a more sensitive and precise analysis of the DES, being the DSC and NMR approaches widely used and accepted and therefore these techniques were performed in this study. The authors were able to successfully verify the establishment of a DES using POH and IBU. As such being that the in-depth analysis of the intermolecular interactions and supramolecular structure of the formulated THEDES is not the main objective of this study, the authors do not feel the need to include FT-IR data in the current manuscript.
- In the data of NMR, the range of X-axis should be unified (use same range in each NMR data)
X-range of the NMR was unified as best as possible to display all relevant peaks (Refer to Results 3.1, L206-212; Figure 3 and 4)
- Table 2, what is the difference between NI and 0.00?
The authors thank the reviewer for the observation. In practical terms, they signify the same namely lack of inhibition, with one “nomenclature” having been applied to the negative control (sterile water) and the other to the various formulations tested. As per the reviewer’s recommendation and to avoid confusion, in the revised manuscript only the designation NI was applied to signify lack of inhibition.
- Table 5, control sample must be necessary to clarify the benefits of present systems.
The authors thanks the reviewer for the comment. The requested data has been added to Table 5 (refer to Results section, 4.3 THEDES anticancer activity, L281-285, Table 5).
References
- Pereira, C.V.; Silva, J.M.; Rodrigues, L.; Reis, R.L.; Paiva, A.; Duarte, A.R.C.; Matias, A. Unveil the Anticancer Potential of Limomene Based Therapeutic Deep Eutectic Solvents. Scientific reports 2019, 9, 1-11.
Reviewer 2 Report
I am satisfied with the changes made to the manuscript by the authors. So, I recommend publication of this article in Pharmaceutics.
Author Response
Reviewer 2
- I am satisfied with the changes made to the manuscript by the authors. So, I recommend publication of this article in Pharmaceutics.
The authors would like to thank the reviewer for his time and contributions towards improving the quality of the manuscript.
Reviewer 3 Report
The way authors revised the manuscript is very difficult to follow. In the future, avoid this kind of editing process.
Anti-microbial activity section:
There is no related discussion about gentamicin and fluconazole study with prepared eutectic formulations. this comparioson analysis should be included in the manuscript.
And authors are misguding the reviewer without explanation of deleted solubility section. "At the same time authors written replied to reviewer comment saying that the solublity section has been modiifed as per reviewer suggesttion". See below regarding this point
Reviewer comment: It seems antimicrobial activity of obtained formulations similar activity when compared to POH. However, authors need to explain in which route of administration these formulations are potential. As well, dissolution measurements in phosphate buffer saline media for these formulations and control IBU should be required which will give an idea of how much solubility advantage.”
Authors answer: The authors understand the reviewer’s point. As per the reviewer’s recommendation solubility studies regarding the THEDES and its parent compounds were performed and included in the revised manuscript.
This is very uncommon.
Author Response
Reviewer 3
- The way authors revised the manuscript is very difficult to follow. In the future, avoid this kind of editing process.
The authors would firstly like to apologize for any confusion caused to the reviewer. Due to a reason alien to the authors, the manuscript made available for revision marked the bulk of the modifications made to manuscript as “deleted”. In this letter, the authors will address the reviewer’s concerns point-by-point emphasizing the necessary parts on the revised manuscript. The authors will know address each point in separate stating the Line and nature of the change done in the manuscript (These changes will also be highlighted in the resubmitted manuscript that will accompany this rebuttal letter).
- Anti-microbial activity section: There is no related discussion about gentamicin and fluconazole study with prepared eutectic formulations. this comparison analysis should be included in the manuscript.
The authors thank the reviewer for his pertinent observation. The applied antibiotic gentamicin and antimycotic fluconazole were included in this study only as an internal control to verify normal microbial strain behaviour. It was not the authors aim in this work to carry out a direct comparison of the formulations with antibiotic/antimycotic compounds, but rather only highlight the fact that the develop formulations possess antimicrobial activity. In fact, to carry out a current comparison analysis of our formulations with gentamicin or fluconazole additional assays (e.g. MIC/MBC determination; Cytotoxicity against specific cell lines) would need to be performed. To avoid causing confusion to the reader, the authors removed the table entry corresponding to gentamicin and fluconazole (refer to Result section 3.3, Table 2, L244-248)
- And authors are misguiding the reviewer without explanation of deleted solubility section. "At the same time authors written replied to reviewer comment saying that the solubility section has been modified as per reviewer suggestion". See below regarding this point
Reviewer comment: It seems antimicrobial activity of obtained formulations similar activity when compared to POH. However, authors need to explain in which route of administration these formulations are potential. As well, dissolution measurements in phosphate buffer saline media for these formulations and control IBU should be required which will give an idea of how much solubility advantage.”
Authors answer: The authors understand the reviewer’s point. As per the reviewer’s recommendation solubility studies regarding the THEDES and its parent compounds were performed and included in the revised manuscript.
This is very uncommon.
The authors once again apologize to the reviewer. As mentioned above, the majority of the modifications made to the manuscript were marked as deleted. The authors did not intend for this to happen and did not wish, in any way, to misled the reviewer. In this version of the manuscript the section regarding solubility has been correctly included (Refer to Result section 3.2 - Solubility of ibuprofen in in a physiological simulated media; L226-236)
Round 2
Reviewer 1 Report
The authors revised the manuscript appropriately as per the reviewers comments and suggestions.